# Advances in the Current Understanding of How Low-Dose Radiation Affects the Cell Cycle

**DOI:** 10.3390/cells11030356

**Published:** 2022-01-21

**Authors:** Md Gulam Musawwir Khan, Yi Wang

**Affiliations:** 1Radiobiology and Health, Canadian Nuclear Laboratories (CNL), Chalk River, ON K0J 1J0, Canada; Md.Gulam.Musawwir.Khan@USherbrooke.ca; 2Department of Biochemistry, Microbiology and Immunology, Faculty of Medicine, University of Ottawa, Ottawa, ON K1H 8M5, Canada

**Keywords:** LDIR, cell cycle, hormesis, cancer, p21^Waf1^(CDKN1A)

## Abstract

Cells exposed to ionizing radiation undergo a series of complex responses, including DNA damage, reproductive cell death, and altered proliferation states, which are all linked to cell cycle dynamics. For many years, a great deal of research has been conducted on cell cycle checkpoints and their regulators in mammalian cells in response to high-dose exposures to ionizing radiation. However, it is unclear how low-dose ionizing radiation (LDIR) regulates the cell cycle progression. A growing body of evidence demonstrates that LDIR may have profound effects on cellular functions. In this review, we summarize the current understanding of how LDIR (of up to 200 mGy) regulates the cell cycle and cell-cycle-associated proteins in various cellular settings. In light of current findings, we also illustrate the conceptual function and possible dichotomous role of p21^Waf1^, a transcriptional target of p53, in response to LDIR.

## 1. Introduction

It is generally accepted that ionizing radiation is harmful to living organisms, including humans, particularly at high doses [1]. Exposure to ionizing radiation can lead to two broad categories of adverse health effects: Deterministic effects that occur above a threshold dose and whose severity is dose related (e.g., skin reaction), and stochastic effects that have neither a threshold dose nor their severity is dose related (e.g., cancer). However, the probability of incidence of either effect increases with dose [1].

The use of ionizing radiation in the treatment of cancer began shortly after the discovery of X-rays in 1895. In 1896, Emil Grubbe used X-rays to treat a recurrent carcinoma of the breast [2]. Currently, controlled utilization of HDIR is a standard option in treating 20–60% of all new cancer cases [3,4]. However, in the process, normal tissue toxicity, followed by the emergence of second cancers, may arise due to the genotoxic properties of HDIR. Permanent changes in the coding sequence of essential genes may lead to a cascade of events associated with the neoplastic transformation of normal tissue that is unavoidably exposed.

HDIR has well-documented and evident impacts on biological processes, molecular pathways, and cellular functions, but the effect of LDIR on human health remains unclear. Human beings are inevitably exposed throughout their lives to low doses of radiation from natural sources (such as cosmic rays and radon gas) as well as human-made sources. A clear understanding of how LDIR impacts cellular processes and molecular mechanisms is becoming increasingly important, as LDIR is frequently used in research, industrial products, security, and modern medicine. Particularly, per-capita medical radiation exposure has been on the rise at an alarming rate over the last few decades, to the point where it is now roughly equivalent to natural background radiation exposure [5,6].

A growing body of evidence supports that LDIR exposure results in distinct molecular-, cellular-, and tissue-level responses when compared with those observed after HDIR exposure [7]. In contrast to HDIR, which causes numerous alterations to macromolecules, including DNA/RNA damage, robust modulation of cell signaling pathways, and degenerative/carcinogenic effects, LDIR may promote hormesis. Radiation-induced homeostasis, often termed hormesis, is a theoretical concept that suggests that exposure to LDIR stimulates beneficial pathways. Several reports indicate that the hormetic process boosts cell survival and growth, improves immune functions, and enhances cytogenetic protection [8,9]. Most population-based epidemiological studies do not show a threshold for cancer incidence; however, at LDIR doses below 100 mGy, there is uncertainty as to whether significant increases in cancer incidence occur in humans. Intriguingly, accumulating experimental evidence indicates that the linear no-threshold (LNT) hypothesis cannot be supported by the biological findings following LDIR exposure, demonstrating that LDIR exposure rather reduces the risk of spontaneous cancer [10,11,12]. Even though the phenomenon is not universal, multiple cellular and molecular data support the notion that LDIR-mediated adaptive responses induce hormesis and cell proliferation in normal cells, but not in cancer cells [13,14,15]. A half-century-old study first described how normal cells respond differently to LDIR exposure than cancerous cells of the same species; J.B. Little showed in 1968 that normal cells exhibited a transient G1 arrest after 100 mGy, implying that delaying in the cell cycle following LDIR exposure commences DNA synthesis machinery for subsequent cell proliferation [16]. Another study demonstrated that the downregulation of several cell-cycle-regulated genes occurs in normal human fibroblasts even at 100 mGy exposure of X-rays and cesium-137 γ-rays, which occurs in a p53-dependent manner and requires its effector p21^Waf1^ [17]. Coincidently, exposure of the same cells to doses ≤ 100 mGy of γ-rays upregulates the level of a protein (TCTP) involved in DNA repair [18].

The cell cycle pathway is one of the pivotal pathways that has been intricately connected with the cellular responses to radiation for many years. HDIR negatively impacts the progression of the cell cycle, and irreparable DNA damage caused by radiation causes the cycle to stall [19,20,21,22]. Even though there have been recent breakthroughs in understanding the molecular mechanisms of the cellular responses to LDIR, its role in the cell cycle arrests/enhanced progression remains unclear in different cellular contexts. There is increasing evidence that LDIR activates multiple signaling pathways to promote cell proliferation (e.g., activation of the Raf, AKT pathways) [23,24]. Through an intricate communication between DNA damage and cell cycle checkpoints, LDIR can confer radioprotection to normal cells as part of the adaptive response [25]. Several cell cycle regulators have been implicated in adaptive response following LDIR exposure via distinct mechanisms; for instance, in human keratinocytes, LDIR triggers cyclin D1 accumulation in the cytoplasm and regulates apoptosis [26]. The cell cycle arrest caused by LDR exposure was observed in human lymphoblast cells as an adaptive response that requires wild-type p53 [27]. The expression of p21^Waf1^, a cyclin-dependent kinase inhibitor that is regulated by p53, has also been reported to increase in human U397 cells and normal breast fibroblasts following LDIR exposure [28,29]. Another study confirmed transient and permanent G1 cell cycle arrest in human fibroblast populations after LDIR exposure (alpha particles) at a dose of 10 mGy, which was accompanied by increased p53 and p21^Waf1^ expression [30].

Increasing numbers of studies have been conducted to understand how LDIR induces hormesis, adaptive responses to subsequent challenge exposures, radioresistance, bystander effects, and genomic instability in various cellular and radiation exposure contexts. Our goal is to summarize recent advances related to the role of LDIR in cell cycle regulation. This review will also discuss how different cell cycle regulators are modulated by LDIR exposure in normal, cancerous, and stem cells. One of the major problems in LDIR research is that arbitrary exposure ranges are used on different cells and in different studies, making it challenging to extrapolate consistent outcomes. Therefore, this review has considered exposure of up to 200 mGy as LDIR to analyze cell cycle effects. Furthermore, a new conceptual mechanism is proposed to explain how LDIR differentially regulates the nucleocytoplasmic shuttling of p21, a key cell cycle regulator, when compared with HDIR.

## 2. Molecular Mechanisms of Low-Dose Ionizing Radiation

### 2.1. Controversy over the Linear No-Threshold Model

The linear no-threshold (LNT) model is commonly used to estimate the cancer risk assessment caused by ionizing radiation exposure and is based on extensive studies of the Japanese atomic bomb survivors. According to the LNT model, the radiation-induced radiological risk for cancer follows a linear relationship with no threshold between absorbed radiation dose and the incidence of cancer [31]. Several, mostly epidemiological, studies using HDIR and LDIR (below 100 mGy) have endorsed the LNT model to predict the risk of cancer or other radiological diseases [32,33,34]. An increasing number of experimental and epidemiological studies highlight inherent uncertainty in the LNT model, particularly at lower dose points, which raises controversy regarding the validity of this model [11,35,36]. In fact, nonlinear patterns are evident at low levels of radiation, demonstrating deterministic beneficial effects via activation of protective mechanisms that defend against disease, also known as hormesis [37,38,39,40].

### 2.2. Radiation Hormesis

Despite extensive studies on the chemical hormesis model, a specific radiation hormesis dose–response model is necessary for a better understanding of the cellular response to LDIR. Growing scientific evidence suggests that radiation hormesis caused by LDIR confers beneficial effects that outweigh any harmful, thereby reducing the risk of cancer [10,41,42]. Experimental evidence suggests that hormesis induced by LDIR provides a protective effect by increasing the proliferation of normal cells and stem cells, subsequent to a transient cell cycle arrest [10,36,37,38]. Many signaling pathways are activated in LDIR-induced hormesis, including Raf, AKT, ERK MAPK, and Wnt, which promote cell proliferation [14,23,43,44]. LDIR induces hormesis by triggering the DNA-damage repair mechanism and an augmented antioxidant response against reactive oxygen species (ROS), which protects chromosomes from mutations, potentially preventing neoplastic transformation [9,12,45,46]. Experimental evidence demonstrates that LDIR induced hormesis enhances innate immunity (through an increased abundance of dendritic cells, natural killer cells, and macrophage cells) and adaptive immunity (through CD4+ T cells, CD8+ T cells, and regulatory T cells), which helps to combat cancer [47,48,49,50]. Furthermore, LDIR-induced hormesis was also associated with significant modulations of cytokines or chemokines production; for example, an increase in stimulatory cytokines and a decrease in immunosuppressive cytokines promote the proliferation of immune cells and confers anticancer immunity [49]. Figure 1 summarizes our current knowledge of LDIR-induced hormesis. Even so, how the cell cycle modulation in the context of LDIR confers hormesis remains unclear and, therefore, needs to be further investigated.

### 2.3. Biological Responses Associated with Low-Dose Ionizing Radiation

To better comprehend the potential beneficial or harmful impacts of LDIR on human health, radiation researchers have been increasingly focusing on several radiation responses such as adaptive responses, bystander effects, hypersensitivity, radioresistance, and genomic instability [51].

In adaptive response, a low priming radiation dose can stimulate intrinsic stress response mechanisms that allow cells to protect themselves from subsequent higher doses of radiation. In adaptive response, various cellular events occur, including activation of multiple signaling pathways, augmented DNA-damage response, increased antioxidant function, enhanced antiapoptotic function, modulation of mitochondrial function, and cell cycle regulation [52,53]. Experimental evidence confirmed that a number of cell cycle regulators, including CDK2, cyclin E, and p53, play a crucial role in mediating the radio-adaptive response [54].

Nonirradiated or nontargeted cells may develop biological effects due to the signal transmitted from adjacent radiation-exposed cells, which is called a bystander response. Multiple signaling pathways (NF-κB/NADPH oxidase/TGFβ pathways), gap junctional intercellular communication, cytokine release, ROS/RNS or nitric oxide (NO) production, and oxidized cell-free DNA play significant roles in the bystander responses [52,55]. Different cellular contexts (e.g., state of the cellular redox environment), radiation type (sparsely or densely ionizing), and levels of absorbed radiation determine the consequences of a bystander response—namely, whether it is beneficial or detrimental [51]. While most bystander effects resulting in adverse outcomes have been studied mainly with low fluences of densely ionizing radiations (α particles, high atomic number (Z) and high energy (E) HZE particles, as well as exposure to high doses of sparsely ionizing radiations (X-rays, γ-rays), beneficial bystander effects have been observed in several studies with low-dose exposures to sparsely ionizing radiations, with to the use of LDIR in anticancer treatments [55,56,57]. LDIR-mediated bystander effects on the regulation of cell cycle in nontargeted cells would be an engaging research topic in the future. Preliminary evidence shows the relevance of this topic to both radiotherapy and radiation risk assessment [58].

Low-dose hyper-radiosensitivity (HRS) and increased radioresistance (IRR) are two phenomena that are associated with LDIR-induced biological responses. The in vitro experiments show that even very small doses of radiation (<100 mGy) can trigger HRS, but as the dose increases (>300 mGy), most cells will begin to display radioresistance until the dose reaches 1 Gy [59]; this phenomenon inherent to the T-N-PR model of radiation response described by John Calkins in 1973 to help understand how high resistance of living organisms is achieved in an environment where they are erratically or chronically exposed to injurious levels of irradiation [60]. Low-dose HRS is often linked to the impairment of DNA repair mechanisms, NO-mediated cell death, and premitotic cell cycle checkpoints; in particular, cells in the G2 phase of the cell cycle exhibit a stronger HRS effect [51,61,62]. Radiation exposure can activate multiple signaling pathways (JAK2/STAT3/AKT/ERK/JNK) and transcription factors (i.e., p53), which can modulate the radiosensitivity of cancer cells [63,64]. Since the description of the HRS phenomenon, there has been a growing interest in fractionated exposure to multiple LDIR and in low-dose-rate radiation due to its stimulation of system responses that lead to the more effective killing of radioresistant cancer cells [65,66,67]. The association between the radiation-induced arrest of G2 phase cells entering mitotic phase prematurely and low-dose HRS has been confirmed by experimental evidence [61]; however, the regulation of cell cycle in the context of HRS, IRR, and low-dose-rate radiation is poorly understood and requires further investigation at different cellular settings.

Recent studies have brought considerable attention to the effects of LDIR on genomic stability. In vitro LDIR exposure leads to chromosomal aberrations of lymphocytes in the human peripheral blood [68]. Another recent study demonstrated an interesting connection between the cell cycle and genomic instability caused by LDIR. The fractionated LDIR exposure can lead to persistent ROS accumulation in mitochondria and can disrupt the AKT/cyclin D1 signaling pathway. Subsequently, the nuclear accumulation of cyclin D1 can result in cell cycle retardation and genomic instability [69,70]. Based on experimental and epidemiological evidence, a recent metanalytic study attempted to estimate the lowest radiation dose that would lead to molecular changes; in this study, chromosomal aberration in cells began in a range of 1 mGy to 500 mGy and in animal models between 50 mGy and 100 mGy; chromosomal aberration has also been observed in children shortly after computed tomography scans with LDIR exposures less than 200 mGy [71].

## 3. Cell Cycle and Radiation

The cell cycle is a controlled process involving a complex network of regulatory mechanisms with appropriate checkpoints that contributes to cell growth, proliferation, and reproduction. There are four primary phases of the cell cycle: G1 (preparatory phase for division), S (chromosome replication), G2 (preparatory phase for mitosis), and M (mitosis, when chromosomes are distributed to two progeny cells) [72]. The cell cycle process is governed by a number of regulatory proteins, ensuring unidirectional and synchronized progression, which includes cyclin family proteins, cyclin-dependent kinases (CDKs), retinoblastoma protein, transcription factors (e.g., E2F), CDK inhibitors (e.g., p16^INK4^ and p21^Waf1^), CDC25 isoforms, p53 family proteins, and MDM2 [73]. It has been well established that radiation (regardless of type) disrupts the regular course of the cell cycle in normal cells, causing affected cells to stop at a checkpoint during the cell cycle. Radiation-induced DNA damage triggers the activation of G1/S, G2/M, and intra-S cell cycle checkpoints, consequently slowing the progress of radiation-exposed cells in the cell cycle [74]. Radiation-induced DNA damage is sensed by ATM/ATR kinases, whose downstream action initiates the DNA damage response and cell cycle arrest by activating the p53 pathway and its target proteins (i.e., p21^Waf1^) [75,76]. Contrary to what is observed in normal cells, extensive in vitro work by Nagasawa et al. showed in various cancer cell types that exposure to radiation exposure does not induce G1 arrests in tumor cells regardless of the presence or absence of functional p53 [77]. In contrast, other studies showed that such arrests exist and occur in a P53-dependent mechanism, particularly in myeloid malignancies (e.g., lymphoma and myeloblastoma) in a p53-dependent mechanism [78,79,80,81]. Clearly, additional research is required to elucidate the role of the microenvironment in which the experiments are performed. Notably, research that addresses the role of cancer-associated fibroblasts (CAFs) in the induction of G1 arrests in cancer cells will be of particular interest. In this context, recent studies have shown that the presence of CAFs alongside cancer cells greatly contributes to the radioresponse of cancer cells [82].

Exposure of normal cells to radiation results in interruption of the G1/S transition, thereby halting further advancement into the S-phase progression, which allows more time to repair DNA damage prior to DNA replication. The arrest is often transient but can become permanent following exposure to HDIR. When double-stranded DNA breaks occur, a G2/M arrest occurs, which prevents cells from entering the M (mitosis) phase, and this cell cycle arrest in G2 allows the coordinated repair of the damage [83]. In severe cases of radiation, the recovery process can be delayed, and sometimes, irreparable DNA damage can lead to mitotic catastrophes that cause cell death [84]. It has been extensively discussed how HDIR modulates the cell cycle, but too little attention has been devoted to how LDIR affects the cell cycle, as discussed in the next section.

## 4. Effect of Low-Dose Ionizing Radiation on the Cell Cycle

Exposure of normal tissue to HDIR (>0.5 Gy) can induce persistent perturbations in molecular and cellular functions, which can lead to adverse effects on health. By contrast, the effects of LDIR have not been thoroughly explored in the context of fundamental cellular pathways/signaling, biological process, and molecular functions. Especially, the impact of LDIR on the cell cycle remains ambiguous, and it needs to be examined thoroughly in both cancer cells and noncancerous cells. Here, we summarize the evidence on how LDIR (<0.2 Gy) can regulate the cell cycle in different cell types, ranging from cancerous cells to normal cells and stem cells, and evaluate the clinical potential of these observations.

### 4.1. Cancer Cells

One of the earliest studies that examined the cell cycle response) to LDIR exposure in a human myeloid tumor cell line (ML-1 was published in 2002 and a microarray profile of cDNA was used to analyze global transcriptional response in this study [85]. This study revealed that LDIR exposure to 20 mGy and 50 mGy of γ-rays resulted in the overexpression of *CDKN1A* (encoding p21^Waf1^) and *GADD45A* (encoding GADD45 alpha protein) genes without any detectable increase in apoptosis and showed that such an LDIR dose range is sufficient to induce a transient cell cycle arrest at the S phase. A study conducted concurrently in the same ML-1 cells also noted that low doses of radiation also resulted in rapid induction of *CDKN1A* and *GADD45A* mRNA [86]. It is noteworthy that cell cycle inhibitor p21^Waf1^, a major transcriptional target of p53, commonly inhibits cyclin/CDK complexes [87,88]. In addition, *GADD45A*, a stress response gene, is also known to be involved in the regulation of the cell cycle [89].

The effect of LDIR on modulating the cell cycle in different cancer cells has been poorly studied compared to the effect of high doses. In a papillary thyroid carcinoma model, the effects of low-dose X-ray irradiation were examined with wild-type p53 (TPC-1) and mutated p53 (BCPAP) cells [90]. TPC-1 cells treated with an LDIR dose of 62.5 mGy of X-ray showed significant decreases in the fraction of cells in the S phase of the cell cycle, along with a concomitant upregulation of p16; however, no changes in cell cycle distribution were observed in BCPAP cells in response to LDIR.

Human prostate cells respond to LDIR via activation of the ATM/p53/p21 axis [91]. Prostate cancer cells PC-3 lacking functional p53 were observed to exhibit a significant S and G2/M phase arrest following 75 mGy exposure, whereas normal prostate cells RWPE-1 did not show detectable changes in cell cycle distribution. In the absence of functioning p53, the ATM/p21 pathway is activated in a p53-independent manner, providing an insight into radiotherapy treatment of prostate cancer [91].

Contrary to the findings in prostate cancer, p53-mutated breast cancer cell MDA-MB-231 responded differently upon LDIR exposure. A dose of 150 mGy significantly increased the growth of MDA-MB-231 cells and accelerated their entry into the S phase of the cell cycle, whereas the growth of Hs578Bst cells (harboring wild-type p53) remained unaltered [28]. Accordingly, LDIR exposure activates cyclin-dependent kinases with increased CDK4, CDK6, and cyclin D1 expression, along with a decreased expression of p21^Waf1^ [28].

The role of p53 in DNA damage-induced arrest in G2 is unclear. In DT40 B-lymphoma cells that lack functional p53, LDIR (100 mGy) induced a Chk2-dependent G2 cell cycle arrest that prevented mitotic entry with damaged DNA [92]. Therefore, LDIR-induced cell cycle regulation is strongly influenced by the status of the p53 gene in cancer cells. Contrary to this notion, the capacity of p53 to mediate a radiation-induced G1 arrest in several human tumor cell lines is disputed by several lines of evidence [77,93]. A previous study investigated the association between low-dose hyper-radiosensitivity and activation of a novel arrest checkpoint in the G2 phase [94]. An asynchronous population of Chinese hamster V79 lung fibroblasts and human T98G glioblastoma cells exhibited significant low-dose hyper-radiosensitivity at doses <200 mGy and <300 mGy, respectively, and both displayed delayed mitosis of damaged G2-phase cells in response to LDIR. However, this study does not demonstrate the variability in LDIR thresholds for controlling progression within the cell cycle in different cells, nor does it examine the modulation of cell cycle regulators in the context of LDIR [94].

### 4.2. Normal Cells

LDIR was exploited in another microarray-based gene expression study to determine the biological effects of 100 mGy of X-rays on normal human lymphoblastoid cells (AHH-1) [95]. The results of this study confirm that LDIR exposure substantially modulates a range of signature genes, such as *GADD45A* and *CDKN2A*. Intriguingly, The *CDKN2A* gene encodes two well-known tumor suppressor proteins p16(INK4a) and the p14(ARF) proteins that are capable of controlling the cell cycle [96]. A recent review revealed that p53 is crucial in LDIR-induced hormesis, adaptive response, radioresistance, and genomic stability [52]. Nevertheless, it remains unclear how LDIR exposure modulates the dynamics of p53 transcription in different cellular contexts. There is increasing evidence that LDIR triggers the expression of CDKN1A [97], which is one of the major transcriptional targets of the p53 gene. Furthermore, p21^Waf1^ can also be regulated independently of p53 through other transcription factors as well as kinases [98].

Another study found that 100 mGy X-ray exposure stimulated cyclin D1 expression in HK-18 human keratinocytes, which is another important regulator of the cell cycle upon DNA damage [26]. However, that study found no detectable difference in cell cycle arrest between cells exposed to LDIR and those subsequently exposed to high dose radiation. Therefore, cyclin D1 does not affect cell cycle regulation in human keratinocytes following LDIR exposure; however, it might play a role in LDIR-induced adaptive radioresistance. A recent review discussed the roles of cyclin D1/CDK4 and cyclin B1/CDK1 in LDIR-induced adaptive responses as well as in the modulation of the mitochondrial signaling network in response to LDIR-induced DNA damage to coordinate cellular responses [70].

As conventional two-dimensional monolayer cell culture models do not replicate the complexity of the human tissue, researchers have expanded their study to three-dimensional (3D) tissue models to gain more insights into LDIR response. In a 3D skin model exposed to 0.1 Gy of LDIR or 1 Gy of HDIR (X-rays) and with subsequent post-radiation harvesting, RNA samples were analyzed for the global transcriptional response using microarrays [99]. As expected HDIR modulates a larger number of genes compared to LDIR over the course of the study (24 h). Intriguingly, global gene expression analysis revealed that LDIR exposure triggered a greater number of differentially expressed genes within the first 3 h. In response to LDIR, genes regulating cell cycle distribution displayed a different dynamic when compared with high doses. After 3 h post-irradiation, LDIR-exposed cells showed prolonged G1/S checkpoint arrest, supported by upregulation of p21^Waf1^, Rb, and p130. At this point, LDIR significantly modulated eight cell-cycle-related pathways, whereas HDIR modulated only two. The findings of this study suggested that the initial response within 3 h of LDIR exposure was associated with tissue protection and enhanced survival against stress [99].

A recent in vitro study indicates that LDIR exposure of 100 mGy to primary keratinocytes and U937 (lymphoma) cell lines causes cell cycle arrest and impairs protein synthesis; however, the phase of the cell cycle in which the arrest occurs is not discussed [29].

### 4.3. Stem Cells

Hormesis effects of LDIR on rat mesenchymal stem cells (MSCs) have also been assessed in a recent study through the investigation of proliferation and signaling pathway (MEK/ERK) activation [14]. Cell proliferation was significantly augmented in MSCs exposed to LDIR at doses of 50 mGy and 75 mGy, and there was a significant shift from the G1 phase to the S phase in response to 75 mGy X-rays. A panel of human embryonic stem cells (hESCs) was also studied to identify the radiobiological effects of LDIR on human cells [97]. *CDKN1A* gene expression was elevated significantly in only one of the hESCs, and it occurred only at 2 h of LDIR exposure (50 mGy). However, the gene expression profiles for the cell-cycle-related genes did not show any linear dose–response relationship, even at the lowest doses of ionizing radiation exposure. Human MSCs also undergo a brief cell cycle arrest at the G1 phase after LDIR exposure (100 mGy) as part of the adaptive responses [100].

Another study examined the effect of LDIR on neural stem/progenitor cells (NSPCs), which were isolated from the subventricular zone of newborn mouse pups and further analyzed for proliferation, self-renewal, and differentiation [22]. There was no apparent increase in the fraction of dying cells shortly after exposure to 100 mGy of cesium-137 γ-rays, and the number of single cells that formed neurospheres was not significantly different from the control. Upon differentiation, irradiated neural precursors did not differ from those that were not irradiated in their ability to generate neurons, astrocytes, and oligodendrocytes. By contrast, the progression of NSPCs through the cell cycle decreased dramatically after exposure to higher doses of radiation.

## 5. LDIR Might Regulate Nucleocytoplasmic Localization of p21^Waf1^

Carcinogenesis or neoplasia is characterized by a break in the control of the cell cycle [101]. Recently, p21^Waf1^ has gained attention for its crucial role in many fundamental biological processes beyond cell cycle regulation including DNA replication and repair, gene transcription, apoptosis, and cell motility [98,102,103,104]. In contrast to its role as an inhibitor of the cell cycle, p21^Waf1^ expression is found to be correlated positively with tumor aggressiveness in a variety of cancers [98]. Moreover, p53-independent regulation of p21^Waf1^ and its ability to nucleo-cytoplasmic shuttling are other aspects of its dichotomous oncogenic nature. The nuclear p21^Waf1^ favors its cell cycle inhibitory function, whereas the cytoplasmic p21^Waf1^ expression is oncogenic [105,106]. When aberrant signaling is triggered by extrinsic factors (including radiation), p21^Waf1^ might be phosphorylated and phosphorylation causes its localization to the cytoplasm, where it exerts its antiapoptotic or oncogenic properties [98].

The radiation level might have a major impact on the nucleocytoplasmic localization of p21^Waf1^, although it is unclear how this occurs. It is likely that in normal cells HDIR might induce several kinases that phosphorylate p21^Waf1^ and cause its localization to the cytoplasm, where it might promote oncogenesis through various mechanisms (e.g., regulation of the nuclear assembly of the cyclin D1.CDK4 complex or conferring protection against apoptosis) [107,108]. Conversely, LDIR might inhibit phosphorylation of p21^Waf1^ (or promote dephosphorylation of p21^Waf1^), thereby exhibiting its nuclear cell cycle inhibitory or tumor suppressor action, which is beneficial in controlling oncogenesis (Figure 2). To confirm this hypothetical concept, a more systematic and extensive investigation must be conducted to ascertain the LDIR-mediated regulation of p21^Waf1^ in different cellular contexts.

## 6. Future Directions

To assess the risk and benefits of LDIR in humans, ongoing studies are investigating molecular, cellular, and tissue responses in different genetic and environmental backgrounds. It is known that LDIR can modulate many cell cycle regulators in normal, tumor, and stem cells, which might have different implications in a particular context. The coordination of cell cycle activities in response to low doses of radiation must be extensively studied to learn how cells cope with genotoxic stress; such studies could lead to the development of cell-cycle-targeted therapies for many diseases, including cancer. It would also be interesting to compare the transition through cell cycle phases after LDIR exposure with the determination of genes associated with early cell cycle response.

LDIR can induce a number of cell cycle regulators, including p53/p21^Waf1^, which are known to be involved in several cell-intrinsic functions (such as apoptosis and senescence) besides cell cycle inhibition. LDIR-exposed nontransformed cells should also be examined in terms of cell cycle regulation and the key players involved. This would be a promising innovative solution for eliminating precancerous cells from the population through the LDIR-stimulated bystander effects. A recent study demonstrated that LDIR exposure on HPKs propagates effects that induce cell cycle arrest and protein synthesis repression in U937 cells; this bystander effect induced by LDIR in U937 cells is associated with increased expression of p21^Waf1^ via TGFβ signaling and activation of p38 MAPK activation [29]. It is interesting to note that p53 plays an important role in communicating a stress response to neighboring cells, in addition to coordinating intrinsic and extrinsic cell signals [109]. Due to its key role in controlling the cell cycle, additional research on the role of the tumor suppressor p53 in studies of bystander effects in normal and cancer cells is warranted.

Regulation of the cell cycle is connected to many important biological processes, metabolic pathways, various stress responses, DNA damage responses, DNA repair, etc. The differential regulation of cell cycle mechanisms in response to LDIR and HDIR might help us determine LDIR’s beneficial or detrimental effects. LDIR-induced regulation of cell cycle and radioresistance in stem cells are the next frontiers to be explored, which can lead to advances in regenerative medicine and tissue engineering.

The cancer cell cycle study is a complex area to explore due to its heterogeneous mutational nature and the multitude of deregulated signaling pathways behind it. Due to the heterogeneous mutational landscape prevailing in cancers, LDIR may affect cell cycle distribution and multiple cellular processes differently in different cancers. Nevertheless, the advent of modern genomics, transcriptomics, and proteomics technology can provide a deeper understanding of the global expression and modulation of genes and proteins associated with the cell cycle in cancer cells maintained in vitro or in vivo response of tumors to LDIR. Such research would provide us with a more comprehensive picture of the regime of cell cycle regulation in cancer cells and would illustrate the effects of nearly all mutations implicated in cancer following LDIR exposure.

Pulse low-dose-rate irradiation (PLDR) has gained increasing acceptance in recent years as the treatment of choice for recurrent cancers and radiation-resistant cancers [110,111]. As PLDR takes advantage of the phenomenon of HRS, it is less toxic to normal cells. The effects of PLDR on the cell cycle have not been well explored. In a recent study, PLDR has been shown to be effective against radioresistant head and neck squamous cell carcinoma (HNSCC), which showed more low-dose HRS than an isogenic population of non-radioresistant parental HNSCC cells; as a result of PLDR-mediated G2/M arrest, the cell cycle resumed 24 h after irradiation on parental cells but not on radioresistant cells [112]. Therefore, it is vital to map out the landscape of cell cycle regulators in the context of radioresistant and radiosensitive cells as well as normal cells.

Recently, several new tools for cell cycle analysis, such as the fluorescence ubiquitination cell-cycle indicator (FUCCI) system, chromobodies, and Cycletest reagent, have been developed. For example, the FUCCI system is a technology that uses the cell cycle’s phase-specific expression of proteins and their degradation by the ubiquitin–proteasome degradation system to color-coded phases of the cell cycle in real time. The new tools, combined with time-lapse imaging, will allow us to explore the spatiotemporal patterns of cell cycle dynamics after exposure to LDIR.

## Figures and Tables

**Figure 1 cells-11-00356-f001:**
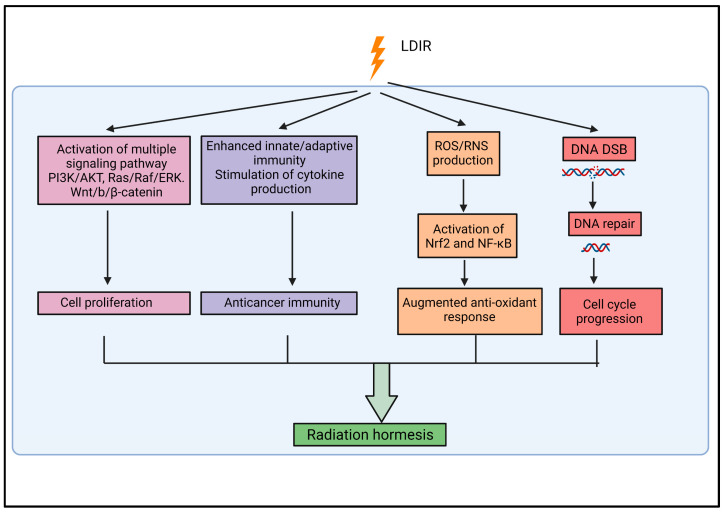
Multiple mechanisms through which LDIR induces hormesis. (RNS, reactive nitrogen species; DSB, double-stranded break).

**Figure 2 cells-11-00356-f002:**
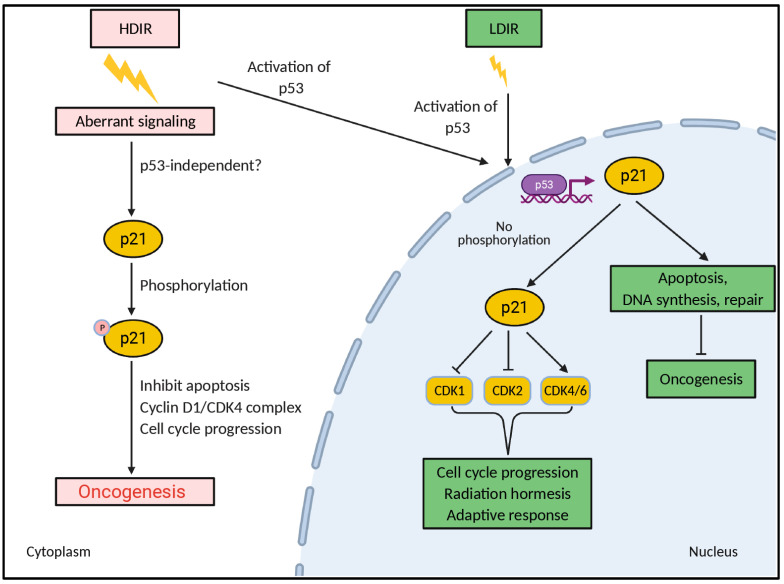
Conceptual diagram showing that radiation dose might differentially regulate the nucleocytoplasmic shuttling of p21^Waf1^. The nuclear function of p21^Waf1^ is predominantly cell cycle inhibitory and tumor suppressive, whereas cytoplasmic p21^Waf1^ can exert antiapoptotic functions. Aberrant cell signaling induced by HDIR might cause phosphorylation of p21^Waf1^ leading to its cytoplasmic localization. Similar to LDIR, HDIR also induces p53-dependent transcription of p21^Waf1^ and p21^Waf1^ nuclear functions.

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
