# Peer review of "Advances in the Current Understanding of How Low-Dose Radiation Affects the Cell Cycle"

_cells, 2022, doi:10.3390/cells11030356_

Round 1

Reviewer 1 Report

Although it is well established that exposure to relatively high levels of ionizing radiation (>0.5 Gy) is harmful, the effects of exposure to low-dose ionizing radiation (LDIR) are less well understood. The present study summarizes the effects of LDIR in the cell cycle and discusses the proteins/mechanisms involved. Because people are commonly exposed to LDIR from numerous sources, understanding how LDIR exposure works may be important to determine its effects on human health. This is a quite comprehensive review on the topic that will be of interest for cell biologists in the field, although I believe it is somewhat lacking in mechanistic detail. Furthermore, some specific issues remain to be addressed before publication:

  1. Line 45. Please explain the term “hormesis” in more detail for clarity.
  2. Line 48. Convert mSv to Gy (100mSv=0.1 Gy) for consistency throughout the manuscript.
  3. Line 75 and line 238. Also, in p53-deficient b-lymphoma cells LDIR induces a Chk2-dependent G2 cell cycle arrest that prevents mitotic entry with damaged DNA (PMID: 17684483). Please discuss.
  4. Line 88: please also state the threshold for LDIR in the abstract for clarity. Please also explain how/why the authors picked this threshold.
  5. Lines 197-199: Do the authors refer to LDIR or to high dose radiation? In high dose radiation, numerous studies report a p53-mediated G1 arrest in cancer cells reviewed in https://doi.org/10.1038/sj.onc.1206677

and references therein. Please clarify/ discuss.

  1. Line 325: “on amino acids, such as serine, threonine, and tyrosine”, redundant, please remove.
  2. Lines 328-330: This is phrased like HDIR promotes anti-apoptotic functions. This is generally not true. However, there are instances where p21 may promote oncogenesis perhaps by promotin assembly of CDK.Cyclin complexes and where loss of p21 delays the onset of lymphoma. This is possibly due to specific genetic nbackground/ cell types PMID: 11751903

PMID: 9405657

Please rephrase.

  1. Figure 2: This figure indicates that high-dose ionizing radiation (HDIR) does not induce p53-mediated-transcription of p21 (unlike LDIR), which is not accurate. Please change accordingly.

Reviewer 2 Report

The effects of low radiation depending of many biological factors. One of them is the cell cycle. In this review the authors discuss some reports showed the effect of P53 mutation on the arrest of cell cycle mediated by p21 in cancer, normal and stem cells. However, the authors must discuss the bystander effect on cell cycle both in tumoral and normal cells, they also must discuss the effect of low radiation on the cell cycle of the radioresistant cells in order to give a landscape of the low radiation in the context of pathologies or healthy.

Authors must show a comparative figure to show the effect of low radiation on the cell cycle in tumor, normal and stem cells.

There are some write mistakes that must be correct. For ex. In figure 1 “Iimunniti”

Round 2

Reviewer 1 Report

The authors successfully addressed my comments, with the possible exception of my previous comment #8. I believe the manuscript is now acceptable for publication with one small final change (please see below):

  1. Figure 2. “HDIR also induces p53-dependent transcription of p21 and p21 nuclear functions (not illustrated in the Figure)”. Please include the above statement in the figure legend of Figure 2.

Author Response

We thank the reviewer for the highly positive comments and constructive suggestions. We have added "HDIR also induces p53-dependent transcription of p21 and p21 nuclear functions" in the figure legend of Figure 2.

Reviewer 2 Report

Authors attended all of my comments. I have not any additional comments

Author Response

We really appreciate the reviewer's suggestions. We would like to thank the reviewer for his time and efforts.